Novel visual analytics approach for chromosome territory analysis

Tkacz Magdalena A. magdalena.tkacz@us.edu.pl 1
Chromiński Kornel 1
Idziak-Helmcke Dominika 2
Robaszkiewicz Ewa 2
1 Faculty of Science and Technology, University of Silesia , Sosnowiec , Poland
2 Faculty of Natural Sciences, University of Silesia in Katowice , Katowice , Poland
Gillespie Joseph
Electronic publication date: 2021 Dec 17
Publication date: 2021
Volume: 9
Electronic Location ID: e12661
Received 2020 Nov 9; Accepted 2021 Nov 30
Copyright: ©2021 Tkacz et al.
Copyright year: 2021
Copyright holder: Tkacz et al.
License: This is an open access article distributed under the terms of the Creative Commons Attribution License, which permits unrestricted use, distribution, reproduction and adaptation in any medium and for any purpose provided that it is properly attributed. For attribution, the original author(s), title, publication source (PeerJ) and either DOI or URL of the article must be cited.
License URL: https://creativecommons.org/licenses/by/4.0/

Keywords: Chromosme territory, Chromosome adjacency, Visualisation, 3D analysis, Computational science, Visual analytics, Nucleus, Nucleus structure, Rice

Funding: The authors received no funding for this work.

==============================
This document presents a new and improved, more intuitive version of a novel method for visually representing the location of objects relative to each other in 3D. The motivation and inspiration for developing this new method came from the necessity for objective chromosome territory (CT) adjacency analysis. The earlier version, Distance Profile Chart (DPC), used octants for 3D orientation. This approach did not provide the best 3D space coverage since space was divided into just eight cones and was not intuitive with regard to orientation in 3D. However, the version presented in this article, called DPC12, allows users to achieve better space coverage during conification since space is now divided into twelve cones. DPC12 is faster than DPC and allows for a more precise determination of the location of objects in 3D. In this article a short introduction about the conification idea is presented. Then we explain how DPC12 is designed and created. After that, we show DPC12 on an instructional dataset to make it easier to understand and demonstrate how they appear and how to read them. Finally, using DPC12 we present an example of an adjacency analysis (AA) using the model of Chromosome Territories (CTs) distribution in the rice nucleus.

Introduction—Motivation and Background

The discovery that the chromosomes occupy distinct areas in the 3D space of the nucleus, so-called chromosome territories (CTs), raised the questions about the factors that determine their mutual positions Cremer et al. (1982). One of the possible approaches to address these questions is to compare experimental data to theoretical models of the nuclei. Our earlier work was dedicated to developing software that would help to model and visualize individual CTs (Tkacz et al., 2016). The resulting Chromosome Territory Modeler (ChroTeMo) and Chromosome Territory Viewer (ChroTeVi) allowed for creating fully probabilistic 3D models of CT distribution in a nucleus and their subsequent comparison to the experimental data obtained by fluorescence in situ hybridization (FISH) with chromosome painting probes Robaszkiewicz et al. (2016). When the CTs are visualized by chromosome painting, they usually display cloud-like shapes with very little overlapping, although it has been proved that at a higher level of resolution CTs intermingle significantly at their borders (Branco & Pombo, 2006). ChroTeMo aims to accurately reflect the images of the nuclei obtained experimentally by FISH and to model the chromosomes with the same level of detail as observed by fluorescence microscopy. Despite the fact that ChroTeMo does not model the inter-chromosomal interactions at the molecular level, nevertheless it allows for assessment of the theoretical frequency of homo- and heterologous chromosome associations, assuming that the chromosome positioning is random (Tkacz et al., 2016). Although the software was originally designed for analysis of the CTs in a model grass Brachypodium distachyon, it can be applied to any plant or animal species, as well as human. Despite the feasibility of modeling algorithm and its application for model visualization, the process of comparing the theoretical and experimental data led to further methodological and algorithmic problems. Although the neighborhood of particular chromosomes or chromosome domains, as well as the distances between them in a modeled nucleus, can be manually assessed and compared by a user to the FISH data, it is a time-consuming process flawed by the analytical subjectivity. In order to address these problems we decided to undertake the task of providing a ChroTeMo and ChroTeVi user with the tools for objective and automated CTs adjacency analysis. The first attempt of using visual analytics for analyzing data that depict real world objects was presented in (Tkacz, 2017). The Distance-Profile Chart (DPC) described in the aforementioned paper is a visualization showing the profile of distances between objects in 3D space. It allows for automatic, reproducible, and objective ways to answer: which CTs (modeled as groups of spheres (see Fig. 1) are associated with each other; what are the distances to the objects surrounding a given CT (e.g., nucleolus, nuclear envelope, other CT s); and how are they positioned in relation to each other.

Figure 1 Visualization of CTs in ricenucleus with the use of ChroTeMo and ChroTeVi (Tkacz et al., 2016).

The gray circular shape represents the nucleus, the yellow sphere represents the nucleolus, beads of spheres represent chromosomes, where one sphere corresponds to a 1 Mb domain. (A) All chromosomes are visible; the ones of interest are in different colors. (B) Only selected chromosomes are visible: chromosome 12′ (labeled blue), chromosome 5′ (labeled orange), chromosome 10 (labeled cyan), chromosome 9 (labeled pink).

Our first attempt used the Hausdorff Distance (HD) for two sets to calculate the distance among CTs. The HD was computed according Eq. (1) (Kuratowski (1980), p.150) (for the two sets, A and B, where x ∈ A, y ∈ B): (1) HdistA,B=maxlimsupdistx,B,limsupdisty,A

However, this approach (using only the HD) gives acceptable results for objects with less complicated shapes than CTs. Since the results obtained using just the HD were not satisfactory for us, we decided to try a different approach.

Inspiration from a shaft of light led us to coin the idea of Cone of Sight (CoS) and the notion of dividing the space into a set of cones that are described below in more detail in the section “Cone Of Sight Idea”. The division of space into a set of cones then allows people to segment 3D space into precisely defined sectors representing directions. The process of dividing 3D space into cones will be referred to as conification. Nevertheless, conification alone does not cover the space entirely as “gaps” between conical surfaces remain.

Unfortunately, we noticed that a relatively large number of points representing a 3D object were not assigned to a cone during conification, which was relatively computationally demanding. Moreover, sometimes this approach did not depict the mutual location of objects well enough. We tried to maximize the 3D space coverage, which led us to the notion of sphere packing. We found that there are two types of sphere packing that maximize 3D space coverage and decided to try to derive cone equations based on an arrangement of spheres.

As a result we obtained 12 cones, which is why we decided to use the abbreviation DPC12 to create the naming convention that describes the number of cones, or more precisely cones of sight (CoS) that were used in the distance profile creation. This paper describes DPC12, which is a modified and expanded version of the DPC method presented in (Tkacz, 2017). The previous paper focused on computational details and was rather a ”proof-of-concept” while this one is the first full journal article concerning DPC s. DPC uses Cartesian Coordinate as the basis for cone creation and space conification. DPC12 uses very intuitive ways of depicting location: front, back, top, bottom, left, and right (Fig. 2). An additional procedure that allows for assigning yet unassigned points to certain CoS is described in this paper later. This paper also contains an additional sample analysis of a CT neighborhood.

Figure 2 Direction naming convention.

Shaft of Light as an Inspiration for the Cone of Sight idea

The idea for the cone of sight was inspired by the shaft of light from a lighthouse (see Fig. 3) or a flashlight beam. When we are in the dark and turn on a flashlight, we point the light in a certain direction, for example, to the left. We can then see objects that are in the shaft of light. When something becomes visible, we can estimate how far it is. In fact, in daylight we use a similar strategy when we look around to recognize and memorize the direction, location, and distance of different objects in the environment.

Figure 3 Shafts of light of the lighthouse spotting different directions.

Photo credit: Evgeni Tcherkasski on Pixabay: https://pixabay.com/pl/photos/latarnia-morska-lekki-morze-2611199/.

Looking around in different directions (left, right, up, down, front, rear) allows us to explore our neighborhood. As a result we are able to make a kind of a “mental image” of our surroundings, a “map” of the objects around. This familiar intuitive approach can be applied to determine the relative location of 3D objects in any space. To use this idea, we will first need to develop a model of a shaft of light. The shaft of light has a shape that is similar to a cone, so it is natural to use it as a model (see Fig. 3). In order to do so, we need to define the description of Cones of Sight in a way that is precise and easy to implement in a computer program, preferably in a form of mathematical equations.

Mathematical model of the cone of sight

The best and easiest mathematical model to describe a shaft of light is a conical surface Encyclopedia of Math (2020), a quadratic surface that can be described with a mathematical equation, thus making it the most suitable solution for computational purposes. The conical surface is defined as a family of straight lines that cross through a certain point, called the cone vertex, and through any of the points on a flat base, which is on a plane that does not contain the vertex. For the purpose of this paper, the base is circular, i.e., it is a circle made by a cross-section of the sphere by the plane, which passes through a diameter (see Fig. 4). An interior limited by and located inside a conical surface could be considered an appropriate model of the interior of a shaft of light. The sectors of 3D space that are limited by the conical surface will be further referred to as Cones of Sight (CoSes).

Figure 4 Conical surface construction: conical surface created by a set of lines n1, n2, .., nk; k ∈ ℕ, crossing through a fixed point O(0, 0, 0) and tangent to a sphere with given coordinates S((a, b, c), R).

Having a set of appropriately arranged spheres we can derive equations for CoS. We wanted to maximize the coverage of space and this led us to the issue of covering space with spheres, which is known as “sphere packing” (WolframAlpha, 2020). This is a known and well-discussed (Conway & Sloane, 1993) problem and will not be covered here in more details. The two types of densest possible space coverage by spheres are hexagonal close packing (HCC) (Weisstein, 2020) and cubic close packing (CCP) (Weisstein, 2021).

To see different sphere arrangements in an interactive way see (WolframAlpha, 2021b) or to observe the difference between HCC and CCP see (WolframAlpha, 2021a). In CCP we have 12 (excluding central) spheres and 12 CoSes. Utilizing the CCP concept in DPC12 we achieved denser space coverage than in DPC, and consequently a more detailed description of the location of surrounding objects. CCP can be easily adjusted to intuitive direction names (Fig. 5).

Figure 5 CoS naming convention.

(A) Sphere arrangement in CCP (WolframAlpha, 2021a). (B) Naming convention for CoSes in different direction based on CCP.

Packing density, which is defined as a fraction of a volume filled by a given collection of spheres (see Eq. (2), Weisstein (2021)) (2) η=VolsphVolunit−cell

is independent of a sphere’s radius and in the case of CCP equals ηCCP=Π32≃0.74048...

which gives us almost 75% of space coverage. The sphere is uniquely defined when we know the coordinates of the center of the sphere and its radius R. To derive the equation of a conical surface, the coordinates of the additional point, which will become the vertex, should also be known.

This point is also assumed to be the geometrical center of the given object (the POV mentioned earlier), in relation to which we want to determine the relative position of other objects. To simplify calculation, it is also assumed that the center of the object is the vertex of the conical surface and is the center of the coordinate system O(0, 0, 0). However, a shift of the center of coordinate system together with all points of the conical surface can be made, if necessary.

The equation of any sphere, with the center in (a, b, c) and the radius R is shown by Eq. (3): (3) x−a2+y−b2+z−c2=R2

For a given sphere (with a center in a point (a, b, c) and a radius R (denoted as S((a, b, c), R)), it is possible to derive the equation of a desirable conical surface (with a vertex in a given point, here in O(0, 0, 0)) in the following way.

The parametric equation of a line in 3D space that crosses a point P0(x0, y0, z0) has the following form: (4) x=x0ty=y0tz=z0t

where t is a parameter.

If point P0 belongs to the line that is also tangent to the sphere, then both equations (of the sphere and the line) have to be fulfilled. So: (5) x0−a2+y0−b2+z0−c2=R2

Substituting the Eq. (4) for the Eq. (5), after making some transformations (according to Stark, 1974, p. 227), we obtain the conical surface equation (Eq. (6)): (6) −2ax−2by−2cz2−4x2+y2+z2a2+b2+c2−R2=0

Changing sign “=” to “≥” in (Eq. (6)) we have an equation that is fulfilled by points that are on the conical surface or inside it.

As mentioned earlier, using spheres with CCP arrangement we have more than the basic six directions shown in Fig. 2. Now we have 12 spheres as shown in Fig. 5A delineating 12 directions with appropriate CoSes. For better readability, we can project the spheres onto a plane as shown in Fig. 5B.

Then, we named directions (and, accordingly CoSes) in the following way (see also Fig. 5B):

• TOP (relative direction: up):

– top left TL –the corresponding direction of CoS is top, but slightly to the left and slightly back;

– top front TF –the corresponding direction of CoS is top, in front;

– top right TR –the corresponding direction of CoS is top, but slightly to the right and slightly back.

• LEFT (relative direction: left):

– left front LF –the corresponding direction of CoS is left, but slightly to the front;

– left L –the corresponding direction of CoS is left;

– left back LB –the corresponding direction of CoS is left, but “over the shoulder” to the back.

• RIGHT (relative direction: right):

– right front RF –the corresponding direction of CoS is right, but slightly to the front;

– right R –the corresponding direction of CoS is right;

– right back RB –the corresponding direction of CoS is right, but “over the shoulder” to the back.

• BOTTOM (relative direction: down):

– bottom left BL –the corresponding direction of CoS is bottom, but slightly to the left and slightly to the front;

– bottom behind BB –the corresponding direction of CoS is bottom, behind;

– bottom right BR –the corresponding direction of CoS is bottom, but slightly to the right and slightly to the front.

To derive the equations of CoSes that correspond to directions it was necessary to determine the centers of spheres in the CCP arrangement. They are presented in Table 1.

Table 1 Coordinates of spheres that are the base for deriving conical surfaces for CoS-es (interactive version with sphere equation available at vailable at Tkacz, 2021).

CoSid	TL	TF	TR	LF	L	LB	RF	R	RB	BL	BB	BR	
a	−1	0	1	−1	−2	−1	1	2	1	−1	0	1	
b	−0.68	1	−0.68	1.68	0	−1.68	−1.68	0	1.68	0.68	−1	0.68	
c	1.68	1.68	1.68	0	0	0	0	0	0	−1.68	−1.68	−1.68	

Efficiency of 3D Space Conification

Spherical, one-group dataset with uniform distribution

As you can see in Fig. 6, CoSes do not fill the space entirely as there remains a little space in between them. This means that after conification, some points still reside outside the 12 CoSes. Each of these points needs to be assigned to the CoSes. To solve this problem, the topological measure(HD), which in this case means the distance from point x to set A (Kuratowski, 1980), p.140)1), was used. See Eq. (7): (7) distx,A= liminf|x−a|,a∈A

Figure 6 CoSes based on conical surfaces derived from the sphere arrangement shown in Fig. 5.

(A) Points assigned to CoS after initial conification. (B) Points assigned to CoS after full conification.

Using HD, any points that remained unassigned after the initial conification process were then assigned to the nearest CoS (Eq. (7)). The conification and assigment of the remaining points to the nearest CoS using HD from here on will be referred to as full conification (FC). After FC, each point Pi in the analyzed set is characterized by five parameters Pi(xi, yi, zi, objid, CoSid). Looking at Fig. 6, we can see the difference between the number of points assigned to CoSes after initial conification without applying HD and the number of points assigned to CoSes after FC.

Table 2 presents the relationship between the conification time and the size of the dataset (number of points subjected to conification). It also shows that the space coverage after initial conification equals at least 80%, which means that less than 20% of the points need to be assigned to the CoSes using computationally demanding HD.

Table 2 Conification time of dataset consisting of one group.

Dataset size [pts]	Time [s]	Conified [%]	
100	0.34	80	
1000	6.34	82.70	
10000	103.8	82.99	
100000	1433.56	83.11	
1000000	10419.12	83.15	

Assigning the sets of points to particular CoSes allows us not only to identify the objects located within a given CoS, but also to determine their nearest and farthest points, as well as to compute the amount of points (e.g., as a percentage) of a given object inside a given CoS. This approach provides us with enough information about objects and their relative location to create a “mental map” of the surrounding space. However, what remains problematic is displaying the obtained, often large amounts of data in a user-friendly way that would enable easy and feasible adjacency analysis. We decided that bar charts will be the most convenient and effective form of visually presenting the resulting data. However, in some applications (e.g., computer vision, machine learning algorithms) storing and processing the data as a table or array can be more useful.

DPC12 in 3D Set Adjacency Analysis

This section describes in detail how to visualize the relative location of 3D objects using DPC12. Because DPC12 was intended as a tool for adjacency analysis of CTs, from here on the analysis of the relative location of objects will be referred to as “Adjacency Analysis” (AA). In the paragraphs below we first describe the process of chart design and composition before explaining how to interpret DPC12 chart using simple, abstract set of objects as an example.

Steps for creating DPC12

The process of creating DPC12 can be described in six simple steps:

1. selecting one fixed object (referred to as central object because it becomes the center of the coordinate system);

2. shifting the center of the coordinate system to the (geometrical) center of central object. This point, now having the coordinates O(0, 0, 0), is the vertex of all cones;

3. making a full conification of the 3D space and assigning all objects to appropriate CoSes;

4. computing what percentage of a given object belongs to a given CoS;

5. computing the distances between the central object and the nearest and farthest points of other objects in each CoS;

6. constructing a visual representation of object locations and distances relative to the central object in the form of a Distance-Profile Chart (DPC12).

DPC12 design

The design of the DPC12 chart was intended to provide the information necessary to create a mental map of all objects surrounding the central object using data computed during steps 4 and 5. During AA, a DPC12 chart can be created for every object in the analyzed scene (space) or only for the objects selected by the user. For example, if there are three objects in the space, up to three DPC12s can be created, one for each object. Using rice nucleus as an example, we could generate twenty-four DPC12 charts, one for each of the twenty-four chromosomes of rice; or we could choose any other number of chromosomes for which we would want to create DPC12s.

The level of detail of the chart depends on the number of spheres from which the CoSes are derived. However, one should take into account that increasing the number of CoSes also increases the complexity of the chart and the difficulty in its reading and interpretation.

The chart consists of a given number of subcharts that correspond to the number of CoSes, so in the case of DPC12 the number of subcharts equals 12. The subcharts are placed on a grid in order to present information in a clear and concise manner, and their names parallel the names of the CoSes (see Fig. 5). The layout of the subcharts of DPC12 is shown in Fig. 7B. The subcharts are patterned after bar plots from which users can read information about the minimum (min) and maximum (max) distances from the central object (Fig. 7A). The color of the bars corresponds to an intensity scale that represents the percentage(fraction) of an object inside a given CoS. The scale used in this article is discrete, made from the sequential palette RColorBrewer Zeileis, Hornik & Murrell (2009) and is shown in Fig. 7C. In the next section we will present a simple example of DPC12 and explain how to interpret it.

Figure 7 DPC12 design for central object.

(A) Subchart of DPC12. (B) Arrangement of the DPC12 subcharts. Each subchart corresponds to a specific CoS. (C) Intensity scale showing the percentage of an object located inside a given CoS.

In summary, we can divide the DPC12 interpretation process into three steps that involve:

• identifying the objects in each CoS and their position (direction) relative to central object;

• determining the distances of these objects from central object and the range of distances they span;

• reading the fraction or percentage of object(s) inside a given CoS (using the color intensity scale).

Instructional reading of DPC12 –artificial 3D Mouse Set

In this section we will demonstrate how to interpret DPC12 using a relatively simple set of objects in 3D space. It consists of three spherical objects positioned in a way that they resemble the head of a mouse with asymmetrically positioned ears (see Fig. 8). We will conveniently refer to this set as a “3D Mouse Set”, abbreviated M3DS.

Figure 8 Mouse 3D set–instructional, briefing set for “how-to” DPC12 reading.

In Fig. 8 we see three objects: the head, abbreviated H, and two ears, left (LE) and right (RE). Due to their fixed positions, they can be rotated as a whole relative to the external coordinate system because such rotation has no impact on AA. In this example the ears are situated above the head. Starting from the left, LE is the first object, H is the second object and RE is the third object.

Since we have three objects, three DPC12s need to be created: one for LE, the second for H, and the third for RE. We need to perform three full conification; each one will feature a difference central object. Thus, the three CoSes will be: LE, H, and RE.

The effects of the three full conifications are shown in Fig. 9. The different colors correspond to different conses oriented in various directions according to Fig. 5. When LE is used as a co, the LE itself occupies several CoSes. In contrast, the entire RE is located in just one CoS, together with a part of H. The rest of H is located in several CoSes (Fig. 9A). By analogy, the same applies to RE, when RE is central object (Fig. 9C). When H is central object, it also occupies a few CoSes, while LE is almost entirely in one CoS and all of RE is in another CoS (Fig. 9B).

Figure 9 Mouse 3D Set conified.

For full resolution images, see GitHub repository (GitHub, 2020). (A) Conified with LE (1st object) as central object. (B) Conified with H (2nd object) as central object. (C) Conified with RE (3rd object) as central object.

When the set is fully conified and the distances in all CoSes for all objects are calculated, we can then construct the DPC12 (chart). The three central objects and their respective DPC12s are shown in Fig. 10. More specifically, DPC12 for LE is shown in Fig. 10A, for H is shown in Fig. 10B and for RE is shown in Fig. 10C. Although the data concerning central object could be removed from the chart, we have decided to display them since it allows us to gain some insight into the features of central object. In all DPC12subcharts within each of the Figs. 10A–10C, the leftmost bar on the horizontal axis of the subchart corresponds to LE, the second (middle) one to H, and the rightmost to RE.

Figure 10 DPC12 for Mouse 3D Set.

For full resolution images, see (GitHub, 2020). (A) DPC12 for LE as central object (AA for LE). (B) DPC12 for H as central object (AA for H). (C) DPC12 for RE as central object (AA for RE).

Analysis for mouse 3D head

We will begin with the AA of H. Let us take a look at the DPC12 with the “head” H as the central object (Fig. 10B). First, we notice that H is present in every CoS. The bars corresponding to H have the lowest min values. They do not start at zero because the center of a coordinate system is placed at the geometrical center of the object. Consequently, in some cases it is possible that the center of a coordinate system might not coincide with any of points comprising the object. It can be seen from DPC12 that the points comprising the “head” form a spherical shape since the range of corresponding bars (bar length) is equal in all CoSes (directions). Also noteworthy, in BB CoS there are fewer head-forming points than in the other CoSes, which is indicated by the lighter bar color. Next, we can see what the location is of the other objects around H. This process mimics how we look around to explore our surroundings. We read from the chart (for the naming convention of directions refer to Fig. 7B the way that the objects are situated in relation to H:

• H (Fig. 9B) exists in every cone (in every direction). This should be apparent since H is now situated in the center of the coordinate system as the CO;

• both ears lie in a range of TOP cones and are situated on opposite sides (TL and TR);

• the first ear lies in a range of the LEFT CoSes, located up and left;

• the second ear lies in a range of the RIGHT CoSes, located up and right;

• there is nothing except H below in the BOTTOM CoSes.

Having oriented ourselves as to the location of each object, we should now analyze the colors of the corresponding bars and the min-max values of the range of the bars in the subcharts. The colors inform us which part of the object is situated within a given CoSes. In the case of the TOP subcharts for H, the color corresponding to the first object (LE) is dark brown. The color of the third object (RE) is also brown, but slightly lighter than the first object. Comparing these colors to the scale presented in Fig. 7C, we can also read that the first object (LE) is almost all (90–100%) present in TL CoS. The rest, up to 10%, is located in the L CoS (very light yellow). In the case of the third object, light brown corresponds to a range of 80–90% for the TR CoS and 10–20% for the R CoS.

Having examined the CoSes in terms of objects situated around H, we can now read the min-max range of their bars, which informs us about how distant these objects are in certain CoSes. We see that the first object (TL CoS) is nearer to H than the third object (TR CoS). Both span approximately the same range, which means that they are probably similar in size. The nearest point of the first object is nearer than the nearest point of the third object, which means that they are not situated symmetrically. Thus, we can state that the first object is closer than the third object.

Analysis for the LE

LE as a central object is obviously present in all CoSes. As was the case with DPC12 for H, its bars do not start at zero. Looking at the bar ranges and colors we can deduce that the points of the second object are not distributed in a uniform way. There are two other objects in only the R, BB, and BR CoSes. That is correct when referring to Fig. 10A: RE is to the right of LE, H is located below the LE and slightly to the right. Looking at R CoS subchart we can see that the min bar value of the third object is higher than the min bar value of the second object, indicating that the third object is farther away from LE than the second one. The third object appears only in R CoS. Additionally, the color of the corresponding bar is dark brown - that means that all points comprising the third object are inside this one CoS. The second object appears in three CoSes (R, BB, and BR), so we know that it is distributed more widely in space than the third object. Looking at bar colors (and bar lenghts) we can infer that most of the second object is in BR CoS (50–60%), next 20–30% is in L CoS and the smallest amount (10–20%) is in BB. As a result we know that there are two objects near LE and they differ in size. One smaller object (RE) is situated just to the right while the bigger object (H) is also to the right but below. The bigger one is nearer to the central object than the small one. You can compare this “mental image” with Fig. 10A.

Analysis for the RE

The analysis and interpretation of DPC12 of RE is very similar to the one described for LE because the M3DS is somewhat symmetrical. However, a careful look at Fig. 8 reveals imperfect symmetry. The “ears” are not equidistant to the “head.” It is possible to determine this asymmetry from looking at the DPC12 alone by comparing the distances in DPC12 created for H. We can detect the asymmetry of the M3DS by reading in Fig. 10B that the distance from central object (H) to LE is longer that the distance between H and RE.

Summary of analysis for M3Ds

By reading DPC12 alone without seeing a picture we can conclude the following. The area consists of three objects in which one is approximately twice the size as the two others. The smaller objects are similar in size and located opposite each other. One is situated farther from the bigger object than the other one. The smaller objects are more distant from each other than they each are from the bigger object. Using the directions set by conification and “pinning” the biggest object at the bottom we can say that the two smaller objects are situated mostly above the bigger one, flanking it on the left and right sides, and one of them is farther away than the other. As has been demonstrated, using DPC12 is scientifically effective and beneficial as DPC12 contains enough information to recreate (after some training and experience in reading the charts) relative object location, distribution, and distances in 3D space without making a 3D visualization. In the next section we will show how to use DPC12 for CT neighborhood (adjacency) analysis.

Adjacency Analysis of RICE Chromosome Territories Using DPC12

In the previous section, we have explained in detail how to read DPC12s using the M3DS dataset as an example. In this section, we show how to use DPC12 in adjacency analysis of selected CTs’ mutual spatial location using ricenucleus (presented in Fig. 1A) as a model. Rice has 12 pairs of chromosomes, so there are twenty-four 3D objects to analyze. Since chromosomes usually have more complicated shapes than the objects analized in M3DS, the ability to properly read DPC12 is very important. For such complicated shapes, the geometrical center would likely reside outside the object. That is why in the case of chromosomes, the center of a coordinate system is located in the centromere instead of the geometrical center. Consequently, placing the center of a coordinate system in the centromere means that the bars for the analyzed chromosome could not start at a zero distance.

DPC12s for all other chromosomes from this model are similar, they are all attached in Supplemental material and can be analyzed similarly as the ones described in sections below.

AA for chromosome 5′ of rice

The chromosome 5′, which is visible in Fig. 1 in orange, is our central object for current analysis. We would like to assess the distance of other chromosomes in relation to 5′. Obviously, the maximum possible distance between two chromosomes cannot exceed the diameter of the nucleus. In the given example of the modeled rice nucleus its diameter equals 7.3 µm.

Looking at Fig. 1 it is easy to notice that the:

• nearest CT, which is pink and represents chromosome 9, is situated on the right side of the chromosome 5′;

• farthest visible chromosome, which is cyan and represents chromosome 10, is situated across from chromosome 5′ on the opposite side of nucleus;

• chromosome 12′, which is blue, is seemingly located farther away from 5′ than chromosome 9 and situated closer to the center of the nucleus.

In order to determine whether we are able to obtain the same information from DPC12 for the chromosome 5′, let us take a look at Fig. 11. We first notice that a few CoSes are almost empty as they contain no or only a small fraction of CTs. These CoSes are:

• TL, TF, RF, R, LF. Based on this, we can deduce that the 5′ chromosome is situated rather close to the nuclear envelope as there are areas around it where no other chromosomes can be found;

• TR, where we can see chromosome 5 and chromosome 5′. The bar’s color intensity indicates that only small portions of their CTs are located in that CoS.

The majority of chromosomes are situated in four CoSes: RB, L, LB, and BB. To determine which chromosomes are the nearest and farthest in relation to 5′, we should compare the minimum values (min) on the y axis of the bars in all subcharts of DPC12 created for 5′. We should also pay attention to the color of the bar.

Identifying the chromosome nearest to rice chromosome 5′

In order to identify the chromosome nearest to 5′, we need to find the chromosome bars with the lowest min value on the y axis. We can see on the BL subchart that min on y axis for chromosome 4′ is about 0.7 µm. However, on the RB subchart the min for chromosome 9 is even lower and equals about 0.5 µm. The bar spans about 2 µm above its min, reaching its maximum value (max) of 2.5 µm. Moreover, the bar for chromosome 9 is the darkest colored, which means that, based on the intensity scale in Fig. 7C, almost all of chromosome 9 is located in the RB. We could then infer that chromosome 9 is the nearest to 5′. This calculation agrees with the model of chromosome territories presented in Fig. 1, where chromosomes 5′ and 9 seem to be tangent. Reading the DPC12 provides not only the same information about the nearest chromosome in an objective way, but also numerical, comparable values of distance. Moreover, using DPC12 does not entail the need to run special software that requires powerful hardware for manipulating 3D objects. Furthermore, we do not have to make manual rotations of the nucleus in order to view the chromosomes of interest, but instead we can read the distances between all of them in one chart.

Figure 11 DPC12 for riceCTs model with the chromosome 5′ as central object.

The ChroTeMo model of the chromosome 5′CT is presented in Fig. 1 in orange.

Figure 12 Visualization of the CTs farthest from chromosome 5′.

Chromosome 5′ (labeled orange) is the central object. Chromosome 9′ (labeled yellow) was identified by DPC12 as the farthest from 5′. Chromosome 10 (labeled cyan) was incorrectly identified as the farthest one after visual human observation.

Identifying the chromosome farthest from rice chromosome 5′

Let us determine which chromosome is the farthest from 5′. The nucleus model in Fig. 1. suggests that it is chromosome 10. In order to confirm this hypothesis, we need to search all the subcharts presented in Fig. 11 for the chromosome bars with the highest bar min value. First, we can notice that there are no bars in any of the subcharts with the min value above 6 µm. On the LB subchart the bar for chromosome 3′ starts at c. 5.2 µm and the bar color indicates that about 60% of 3′ is located in this CoS. On the same subchart, for chromosome 9′, the min bar value and the percentage of CT in LB equal approximately 5.5 µm and 40%, respectively. The bars for the same chromosomes on the BBsubchart start at the value 5.2 µm for chromosome 3′ and 5.6 µm for 9′, and the percentage of chromosome area located in this CoS equals 40% for 3′ and 60% for 9′. In comparison to 3′ and 9′, chromosome 10 appears entirely within BB and its bar min value is c. 4.8 µm, which indicates that it is closer to 5′ than 3′ or 9′. Based on the data provided by DPC12 , we can conclude that chromosome 9′, and not chromosome 10, is the farthest one from 5′. Let us now check our model of rice CTs, how chromosomes 5′, 9′, and 10 are situated in relation to each other. Fig. 12 presents the same model as visible in Fig. 1 after 3D rotation. Its analysis confirms that indeed, chromosome 9′ is slightly farther away from 5′ than chromosome 10, almost on the opposite side of the nucleus across from 5′. This example demonstrates that the DPC12 approach proves to be less error-prone in comparison to subjective, manual human-based assessments of distances between objects in 3D.

Finding the nearest and furthest CT s based on DPC12 for rice chromosome 10

The previous section presented a DPC12 interpretation when the CT arrangement in a 3D model was already known. Now let us see if we can determine the distance between CTs and their spatial distribution based exclusively on DPC12 analysis (when the CT arrangement is not visualized in 3D). For that purpose we will analyze another DPC12 (presented in Fig. 13) in which chromosome 10 shown in Fig. 1 in cyan is the central object. Like in the previous section, we will determine which CTs are the nearest to and farthest from chromosome 10.

Figure 13 DPC12 for rice CTs model with chromosome 10 as central object.

The ChroTeMo model of the chromosome 10 CT is presented in Fig. 1 in cyan.

As we established in the previous section, our initial statement that chromosome 10 is the farthest away from 5′ was wrong, therefore, we can also assume that 5′ is not the farthest from 10. Thus, we must start our analysis from scratch. Relying solely on the values presented in DPC12, like before, we must search for bars that have the lowest and the highest min value on the y-axis in all subcharts. Also like before, we have to take into account the percentage of a given CTsituated in a particular CoS. In the case of DPC12 for chromosome 10, we can see that almost all other chromosomes are situated in the top three CoSes (TL, TF, TR). Hence, we can assume that this chromosome is situated near the nuclear periphery.

Identifying the chromosome nearest to rice chromosome 10

In order to find the chromosomes closest to chromosome 10, we first need to identify the subcharts with the lowest situated bars. In Fig. 13 these are TF, RF, LF, TR, and LB. To facilitate comparison between the bar min values, we added blue baselines that serve as a ruler-guide (see Figs. 14 and 15).

Figure 14 Identifying the CTs nearest to chromosome 10.

Selected subcharts zoomed-in from the Fig. 13. The ChroTeMo model of the chromosome 10 CT is presented in Fig. 1 in cyan.

Figure 15 Identifying the CTs nearest to chromosome 10 (cont.).

Selected subcharts zoomed-in from the Fig. 13. The ChroTeMo model of the chromosome 10 CT is presented in Fig. 1 in cyan.

We will begin by examining the CoSes: TF, RF, and LF (zoomed-in in Fig. 14). In TFthe lowest min value belongs to the bar corresponding to chromosome 4; it equals approximately 0.7 µm. Based on the intensity scale in Fig. 7C, we can determine that about 70% of chromosome 4 is situated in this CoS. In both subcharts RF and LF the lowest min value of the bar also belongs to chromosome 4 and equals approximately 0.9 µm. The weak color intensity of the chromosome 4 bars in both CoSes means that only a small part of chromosome 4 is near chromosome 10, c. 10% in the case of RF and 20% for LF.

Next, we will examine subcharts TR and LB (Fig. 15). In TR the lowest min value of 1.1 µm can be read for chromosome 3′ and 80% of this chromosome is located within the CoS. In LB chromosome 3′ has the lowest min value of 1.3 µm and c. 20% of this chromosome is located in this CoS. The results of the analysis presented above are summarized in Table 3. Comparing the parameters allows us to state that the closest neighbor of chromosome 10 is chromosome 4, while 3′ is slightly farther away, but still very close. In the next section we will show readers how to find the farthest chromosome from chromosome 10.

Table 3 Summary of min bar values in DPC12 and percentage of CT in CoS.

	4	3′	
TF	0.7 µm (70%)	NA	
RF	0.9 µm (10%)	NA	
LF	0.9 µm (20%)	NA	
TR	NA	1.1 µm (80%)	
LB	NA	1.3 µm (20%)	

Identifying the chromosome farthest from rice chromosome 10

Now that we have identified the chromosomes nearest to chromosome 10, we can proceed by analogy to find the ones that are most distant from 10. To do that, we must first select the DPC12subcharts containing the bars with the highest min value. In this case these are TLand TFsubcharts, zoomed-in and presented in Fig. 16.

Figure 16 Identifying the CTs farthest from chromosome 10.

Selected subcharts zoomed-in from the Fig. 13. The ChroTeMo model of the chromosome 10 CT is presented in Fig. 1 in cyan.

We can read in Fig. 16 that in both subcharts the farthest chromosome from chromosome 10 is chromosome 9. The bars indicate that in TLthe chromosome 9 bar spans the distance from 4.9 µm to 6.1 µm, whereas in TFthe same chromosome spans the distance from 4.9 µm to c. 5.8 µm. Up to 40% of chromosome 9 CT is located in TL, and the remaining part of its territory is in TF.

After analyzing DPC12 charts with chromosome 10 as the central object we can definitively state that:

• chromosome 10 is situated near the edge of the nucleus, which is indicated by the almost empty top CoSes (TL, TF, TR)

• no other chromosomes separate chromosome 10 from the nucleus edge, which is indicated by the completely empty bottom CoSes (BL, BB, BR)

• the nearest CTs to chromosome 10 are the CTs of chromosomes 3′ and 4

• the farthest CT from chromosome 10 is the CT of chromosome 9.

We can now compare the data derived from DPC12 analysis to the information provided by the ChroTeMo model of rice nucleus (Fig. 17). As we can see, the conclusions drawn from reading DPC12 are in full agreement with the 3D nucleus model concerning the spatial positioning of chromosomes 10, 3′, 4, and 9. Furthermore, this full agreement proves that DPC12 is sufficient not only for identifying the nearest and farthest CTs in regard to a given chromosome, but also for performing detailed adjacency analyses of CTs without the necessity of visualizing them in 3D.

Figure 17 Visualization of the CTs nearest to and farthest from chromosome 10.

Chromosome 10 (labeled cyan) is the central object. The two nearest chromosomes are 4 (labeled green) and 3′ (labeled gray). The farthest chromosome is 9 (labeled pink).

Conclusions and Future Work

In this paper we introduce a novel, user-friendly approach for assessing the mutual location of 3D objects by using the Cone of Sight (CoS) concept. The advantage of this approach is that it does not require specialized tools for visualization and interaction with visualized 3D objects, which is often computationally demanding and requires powerful hardware. DPC12 analysis of the mutual location of 3D objects provides sets of objective measurements and its results are reproducible and can be easily shared as image or pdf files. Moreover, the DPC12 approach is less time-consuming than manually manipulating objects in 3D because it allows researchers to perform object adjacency analysis using just one chart. The approach can be modified and applied to other research areas. Its design makes it usable in any type of data analysis where 3D objects can be represented or approximated by sets of 3D points or vector features.

Our further plans include:

• optimizing the code

• preparing the code as a module (library) to make it more accessible and user-friendly

• developing the code to allow for the creation of an interactive version of DPC12

• developing an algorithm to convert microscope images to digital models, which will enable using the DPC12 approach to analyze experimental data.

Additional Information and Declarations

Competing Interests

Author Contributions

Data Availability

1 In fact, this is a special case of HD mentioned earlier where set A = {x} consists of one point

The authors declare there are no competing interests.

Magdalena A. Tkacz conceived and designed the experiments, performed the experiments, analyzed the data, prepared figures and/or tables, authored or reviewed drafts of the paper, idea, calculations, deriving equations, and approved the final draft.

Kornel Chromiński conceived and designed the experiments, performed the experiments, analyzed the data, prepared figures and/or tables, authored or reviewed drafts of the paper, developer, and approved the final draft.

Dominika Idziak-Helmcke and Ewa Robaszkiewicz conceived and designed the experiments, analyzed the data, authored or reviewed drafts of the paper, and approved the final draft.

The following information was supplied regarding data availability:

The data is available at GitHub: https://github.com/Kornelch/DPC-visualisation.

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
