# Peer review of "Novel visual analytics approach for chromosome territory analysis"

_PeerJ, doi:10.7717/peerj.12661_

## Round 0.1 · original submission · Major Revisions

Dear Dr. Tkacz and colleagues:

Thanks for submitting your manuscript to PeerJ. I have now received two independent reviews of your work, and as you will see, the reviewers raised some concerns about the research and are not very enthusiastic about your manuscript. Despite this, I encourage you to revise your manuscript, accordingly, taking into account all of the concerns raised by both reviewers.

Perhaps selecting a better data set (per R2) would garner more enthusiasm, or better framing your study on how the novel approach improves the field beyond an incremental advancement.

I look forward to seeing your revision, and thanks again for submitting your work to PeerJ.

Good luck with your revision,

-joe

Reviewer 1 ·

Basic reporting

Generally the English is satisfactory, but there are several errors:
1) The abbreviation "CT" should be defined in the abstract.
2) Figure 3 is not referenced in the main text.

Experimental design

The coordinates in table 1 are either incorrect, or they need better explanation.

Validity of the findings

The manuscript describes a technical advance for visually interpreting the position of objects in 3D space, by dividing the space into 12 cones of sight. The method is an incremental improvement over a previous method, and therefore is only somewhat novel. Impact is limited, because the utilization of the method to generate results more easily than other methods has not been reported. Although, application to chromosome territory analysis is significant, it is very likely that the method is too simple to deal with the complex shape of chromosomes. Chromosome "painting" shows using 3D fluorescence microscopy that each chromosome is "blob like" and they are in full contact with each other. However, at a high spatial resolution, the 3D picture is more complex with chromosomes more intertwined with each other. It is unlikely that the proposed method can deal with such complex interactions.

Reviewer 2 ·

Basic reporting

No comment

Experimental design

The example chosen (3D Mouse Set), although cute, is inappropriate to appreciate the potentiality of the method. The authors should provide a more realistic example, if possible real data (as example the ones of their previous paper: https://doi.org/10.1371/journal.pone.0160303)

Validity of the findings

With the actual 'model dataset' is really difficult to test the validity of the findings and the potentiality of the method. I strongly suggest the authors to provide a more realistic model dataset and if possible to test their method on real data.

Additional comments

Line85: "The cone, as a solid block known from stereometry (solid geometry), cannot be described with the equation." Can the authors clarify this point? Why classical mathematical definition of three-dimensional conics is not appropriate? Maybe, here you refer to to CoS not cone?
Can the author discuss dependency of volume coverage to R? 12 is optimal independently of R?

---

## Round 0.2 · Minor Revisions

Dear Dr. Tkacz and colleagues:

Thanks for revising your manuscript. The reviewers are very satisfied with your revision (as am I). Great! However, there are a few additional concerns to address per reviewer 1. Please attend to these issues ASAP so we may move towards acceptance of your work.

Best,

-joe

Reviewer 1 ·

Basic reporting

The explanation of figure 4 is unclear. The set of lines, n1, n2, n3 etc are not labeled in the image, These lines (dashed) lines, which are tangential to the sphere cross at the origin. They do not cross at P0.

Figure 10: a, b and c are not marked in the figure.

Do not use questions in the text.

Experimental design

The manuscript is much clearer than the original submission and it is now clear that the method, DPC12 provides more quantitative, more complete and in some cases avoids errors that would have occurred in standard 3D visualization.

Validity of the findings

The findings have been carefully validated and explained using a simulation of a mouse's head and ears, and using actual 3D image data about chromosome territories (CT) in rice nuclei.

The manuscript describes a novel method to extract quantitative information from 3D images. This is the main value of the study. The application to CT is a good choice and I am not too concerned that the authors have not applied their method to determine new biological results about CTs.

Reviewer 2 ·

Basic reporting

All fine.

Experimental design

All fine.

Validity of the findings

All fine.

Additional comments

The authors well reply to my points and improved the manuscript as required.

---

## Round 0.3 · accepted · Accept

Dear Dr. Tkacz and colleagues:

Thanks for revising your manuscript based on the concerns raised by the reviewer. I now believe that your manuscript is suitable for publication. Congratulations! I look forward to seeing this work in print, and I anticipate it being an important resource for groups studying chromosome territory visualization and analyses. Thanks again for choosing PeerJ to publish such important work.

Best,

-joe